# Spiking Neural Network Based on Multi-Scale Saliency Fusion for Breast Cancer Detection

**DOI:** 10.3390/e24111543

**Published:** 2022-10-27

**Authors:** Qiang Fu, Hongbin Dong

**Affiliations:** College of Computer Science and Technology, Harbin Engineering University, Harbin 150001, China

**Keywords:** spiking neural network, YOLO, medical image, object detection

## Abstract

Deep neural networks have been successfully applied in the field of image recognition and object detection, and the recognition results are close to or even superior to those from human beings. A deep neural network takes the activation function as the basic unit. It is inferior to the spiking neural network, which takes the spiking neuron model as the basic unit in the aspect of biological interpretability. The spiking neural network is considered as the third-generation artificial neural network, which is event-driven and has low power consumption. It modulates the process of nerve cells from receiving a stimulus to firing spikes. However, it is difficult to train spiking neural network directly due to the non-differentiable spiking neurons. In particular, it is impossible to train a spiking neural network using the back-propagation algorithm directly. Therefore, the application scenarios of spiking neural network are not as extensive as deep neural network, and a spiking neural network is mostly used in simple image classification tasks. This paper proposed a spiking neural network method for the field of object detection based on medical images using the method of converting a deep neural network to spiking neural network. The detection framework relies on the YOLO structure and uses the feature pyramid structure to obtain the multi-scale features of the image. By fusing the high resolution of low-level features and the strong semantic information of high-level features, the detection precision of the network is improved. The proposed method is applied to detect the location and classification of breast lesions with ultrasound and X-ray datasets, and the results are 90.67% and 92.81%, respectively.

## 1. Introduction

The incidence rate of breast cancer ranks first among female malignancies [1]. The symptoms of early breast cancer are not obvious. Advanced breast cancer can cause distant metastasis of cancer cells and multiple organ lesions, which directly threaten the lives of patients. Imaging technology [2,3,4] is widely used in breast cancer screening, as it can directly observe the lesions inside the breast and detect early concealed lesions to help doctors review the images and make a judgment of the nature of the masses.

At present, the review of medical images mainly includes manual review and machine review. Manual review means relying on the traditional computer image processing technology to carry out image digitization, transformation, enhancement, restoration, and reconstruction of the collected image data. Radiologists complete the review of the image according to the observation of the computer-processed image. The manual review mainly depends on the subjective judgment of radiologists. Due to individual differences, different radiologists may give different diagnosis results for the same image, and the same radiologist may even give different diagnosis results for the same image in different states. Compared with the manual review, the machine review can reduce the workload of radiologists and avoid subjective judgment to a certain extent.

The application of computer vision in medical imaging is mainly divided into two categories, namely computer-aided diagnosis (CADx) and computer-aided detection (CADe). The CADx can classify, recognize, and predict diseases [5]. However, treating medical imaging as a classification problem, the task setting is too ambitious and extensive, and it cannot be regarded as solving medical problems. Although it can be regarded as a certain degree of computer-aided diagnosis, the attribution and the interpretability are flawed, and it is not enough for doctors to refer to it all. The CADe is mainly used for the detection of lesions or lesions in the image, which is more realistic when performing medical image analysis [6,7,8].

In recent years, deep learning has completely changed the field of machine learning, especially in the field of computer vision. In this method, the most common way to train a deep artificial neural network (ANN) is to use the back-propagation algorithm. A large number of annotated training samples are needed, but the accuracy is indeed satisfactory, and sometimes even better than with humans. The neurons in the ANN have single, static, and continuous activation values. However, discrete spikes, spike time, and spike rates are used to calculate and transmit information in biological neurons [9]. In addition, there are other substances to calculate and transmit information [10]. As a result, the spiking neural network (SNN) is biologically more realistic than the traditional ANN, and it is the only feasible way to understand how the brain calculates at the level of neuron description. However, training deep SNN is still a challenge. The transfer function of the spiking neuron is usually non-differentiable, which will be an obstacle to use the back-propagation algorithm. The SNN is an effective tool for processing complex spatiotemporal information, which is composed of interconnected spike neurons. However, due to its inherent mechanism, how to design an efficient learning algorithm for SNN and what kind of topology is more effective are still important issues in this research field.

In this work, SNN is proposed to detect breast cancer on two modalities of datasets based on the framework named ‘You Only Look Once (YOLO)’ [11]. A method of converting DNN to SNN is proposed to transfer the backbone network to SNN. The network consists of three parts, i.e., feature pyramid networks, the saliency model, and the backbone network.

The main contributions of this paper are as follows:(1)The method of converting DNN to SNN is proposed for the field of object detection based on medical images using the method of converting DNN to SNN;(2)The feature pyramid structure is employed to obtain the multi-scale features of the image, and the method of fusing the high-resolution of the low-level features and the strong semantic information of the high-level features is employed to improve the detection precision of the SNN;(3)A lesion detection model based on multi-scale saliency fusion is proposed;(4)The first SNN-based breast cancer detection model is proposed.

The rest of the paper is organized as follows. Section 2 provides the related works. The multi-scale saliency fusion model and a method of converting DNN to SNN are presented in Section 3. Section 4 introduces ultrasound and X-ray datasets of breast cancer. The experimental results that demonstrate the performance of the proposed methods under two breast cancer detection tasks are provided in Section 5, and Section 6 concludes this paper.

## 2. Related Works

Interventional therapy of medical imaging has significantly improved the level of early diagnosis of breast cancer. With the application of artificial intelligence in the field of healthcare, researchers use image processing and computer vision technology to design effective intelligent computer-aided detection and diagnosis systems.

Object detection based on medical images can be regarded as the location and classification of multiple lesions. Traditional object detection algorithms are based on manual feature extraction, which slowly improves the detection precision by building complex models and multi-model integration based on basic feature expression [12,13]. Due to the fact that convolutional neural networks (CNNs) can learn the feature representation with strong robustness and certain expression ability, a region with CNN features (R-CNN) [14] model is proposed. The important contribution of R-CNN is to introduce deep learning into object detection. However, when R-CNN sends candidate regions to CNN, CNN needs a fixed input size, so the size of the input image cannot be adjusted arbitrarily. In addition, because candidate regions may often overlap, the method of sending each candidate region to CNN will cause a lot of repeated calculations. To solve these two problems, SPP-Net [15] is proposed. The SPP-Net can solve the problem that the size of the input image cannot be adjusted and, thus, saves a lot of computing time. Based on R-CNN, fast R-CNN is proposed [16]. Compared with the multi-stage training of R-CNN, the training of fast R-CNN is more concise. However, the fast R-CNN needs to use an external algorithm to extract the object candidate box in advance. Therefore, fast R-CNN integrates the steps of extracting target candidate frames into DNN [17]. To meet the real-time requirements of object detection, single-stage real-time object detection is realized by YOLO [11] for the first time. The SSD [18] absorbs the fast detection idea of YOLO, combines the advantages of RPN in fast R-CNN, improves the processing method of multi-scale objects, and achieves faster detection performance than YOLO. To solve the unbalanced distribution of object background data in SSD used as a single-stage target detection algorithm, Refinedet [19] combines the advantages of filtering the background area in a two-stage object detection method, and this paper proposes an anchor refinement module and object detection module, as well as transfer connection block for concatenating them. Indeed, YOLOv3 [20] uses several independent classifiers instead of the softmax function and uses a method similar to the feature pyramid network to make a multi-scale prediction.

Deep learning has made great progress in the field of object recognition. Table 1 shows the summary of deep learning methods. Various DNN-based feature extraction architectures are proposed for breast cancer detection and classification [21,22]. However, unlike deep CNNs, limited work has been performed regarding SNNs in the field of object detection. An SNN is mostly used in image classification tasks. A method is proposed for learning image features with locally connected layers in SNNs using the STDP rule [23]. In this approach, sub-networks compete via inhibitory interactions to learn features from different locations of the input space. Indeed, [24] proposes efficient spatiotemporally compressive spike features and presents a lightweight SNN framework that includes a feature extraction layer to extract such compressive features, while [9] proposes an ensemble SNN for the histopathological image. It is used for an eight-classification work, which includes four types of benign tumors and four types of malignant tumors. To the best of our knowledge, our study is the first SNN for breast cancer detection.

## 3. Methods

In this section, a multi-scale saliency fusion model and a transformation method from DNN to SNN are proposed. In the multi-scale saliency fusion model, as shown in Figure 1, a feature pyramid network is used to obtain multi-scale features, and the attention module fuses the spatial and channel attention mechanisms.

### 3.1. Spiking Neural Networks

In this section, the introduction of SNN and the method of converting DNN to SNN are given. The SNNs are composed of spiking neurons interconnected by synapses. Spiking neurons simulate the information transmission mechanism of biological neurons, as shown in Figure 2. This mimics the process that the ion channel on the cell membrane is opened by neurons receiving stimulation, and then the charged ions inside and outside the cell membrane flow to generate an action potential. When the action potential reaches a certain threshold, an action potential is generated. The action potential is transmitted along the axon to the nerve terminal. Finally, it is transmitted to the postsynaptic neuron through the synapse.

Considering the complexity of network scale and model, a simple leaky integrate-and-fire (LIF) [25] neuron model is used for SNN in this paper. The basic circuit of the LIF model consists of a capacitor and a resistor in parallel. As shown in Figure 3, the driving current can be divided into two parts. It can be calculated as follows:(1)$$I\left(t\right)=C_{m}\frac{dV_{m}}{dt}+\frac{V_{m}}{R_{m}},$$
where $C_{m}$ is the membrane capacitance, $V_{m}$ is the voltage of the membrane, $R_{m}$ is the resistance of membrane, and I(t) is the total current of membrane. Here, τ=RC is the time constant of leakage current, which is calculated as follows:(2)$$\tau \frac{dV_{m}}{dt}=-V_{m}\left(t\right)+RI\left(t\right),$$
When the neuron receives a constant current stimulation and the cell membrane is at a resting potential of 0 mv, that is, $I\left(t\right)=I_{0}$, the membrane potential can be calculated as follows:(3)$$V_{m}\left(t\right)=RI_{0}\left[1-\exp\left(-\frac{t-t^{\left(0\right)}}{\tau }\right)\right],$$
where $t^{\left(0\right)}$ is the firing time of the previous spike. If the value of $V_{m}$ is less than the firing threshold $V_{th}$, no spike is generated; on the contrary, if the value of $V_{m}$ reaches the threshold $V_{th}$, an output spike is generated at $t^{\left(1\right)}$. Therefore, the threshold of spike firing can be calculated as follows:(4)$$V_{th}\left(t\right)=RI_{0}\left[1-\exp\left(-\frac{t^{\left(1\right)}-t^{\left(0\right)}}{\tau }\right)\right],$$

The internal spike time interval i.e., $\Delta T=t^{\left(1\right)}-t^{\left(0\right)}$ can be calculated as follows:(5)$$V_{th}\left(t\right)=RI_{0}\left[1-\exp\left(-\frac{t^{\left(1\right)}-t^{\left(0\right)}}{\tau }\right)\right],$$

The ReLU activation function in DNN is very close to the curve of the spiking neuron model, as shown in Figure 4. Therefore, the DNN can be converted into the SNN. We are able to prove this view theoretically. In this paper, the relationship between the firing frequency f of the first layer of the neural network and the activation in the corresponding ANN are discussed [26].

Suppose the input is constant as $z=V_{th}x\in \left[0,1\right]$. The process of changes in membrane potential *V* with time in SNN can be calculated as follows:(6)$$V_{m}\left(t\right)=V_{m}\left(t-1\right)+z-V_{th}\theta_{t},$$
where $\theta_{t}$ is the output spikes. The average firing rate in T time steps can be obtained by summation of membrane potential. It can be calculated as follows:(7)$$\sum_{t=1}^{T}V_{m}\left(t\right)=\sum_{t=1}^{T}V_{m}\left(t-1\right)+\mathrm{zT}-V_{th}\sum_{t=1}^{T}\theta_{t},$$

Then, move all items containing $V_{m}\left(t\right)$ to the left, and divide both sides by T at the same time, as follows:(8)$$\frac{V_{T}-V_{0}}{T}=z-V_{th}f,$$
(9)$$f=x-\frac{V_{T}-V_{0}}{TV_{th}},$$

Therefore, in the case of an infinite simulation time step, the following is true:(10)f=x,

In the training process, DNN uses batch normalization to normalize the output value to a zero mean value to accelerate the training and convergence. It can be calculated as follows:(11)$$y=\frac{\gamma }{\sigma }\left(x-\mu \right)+\beta ,$$
where *x* is input value, μ and σ are mean and variance, respectively, and γ and β are obtained in the training process.

After training, these transformations can be integrated into the weight vector to maintain the performance of batch normalization. However, there is no need to repeat the normalization calculation for each sample. Therefore, this work refers to the method proposed by [27] to calculate the normalization. It can be calculated as follows:(12)$${\tilde{W}}_{ij}^{l}=\frac{\gamma_{i}^{l}}{\sigma_{i}^{l}}W_{ij}^{l},$$
(13)$${\tilde{b}}_{i}^{l}=\frac{\gamma_{i}^{l}}{\sigma_{i}^{l}}\left(b_{i}^{l}-\mu_{i}^{l}\right)+\beta_{i}^{l},$$

This method does not need to transform the batch normalization layer after transforming the weight of the previous layer. Furthermore, when the batch normalization parameter is integrated into other weights, the loss is reduced.

### 3.2. Multi-Scale Saliency Fusion Model

An image pyramid network uses the same image to construct pyramid features through different scales [28]. Compared with single-scale object detection, the advantage of an image pyramid is that it is possible to obtain different scale feature maps by adjusting the resolution of the image, and then to detect different scale objects. Because the image resolution is different, the size of the object and the semantic information of its features are also different. Pyramid features make up for the loss of semantic information in the process of down-sampling, so its detection precision can be improved to a certain extent.

Although the image pyramid network has a certain improvement effect on the detection precision, its disadvantage is that the large datasets occupy a lot of memory and consume a lot of time, so it has been gradually replaced by the feature pyramid network in the development process of object detection. The feature pyramid network (FPN) can achieve both speed and precision, and greatly improves the performance of object detection by improving multi-scale features with strong semantics. However, before the feature fusion in the FPN stage, there are semantic differences between the features of different network layers. The features of different network layers independently pass through the 1 × 1 convolutional layer, the purpose of which is to reduce the channels of the feature vector. However, there is a huge semantic gap between features of different scales. Due to the inconsistency of semantic information, fusing these features directly will reduce the expressive ability of multi-scale features. Therefore, this paper uses the FPN to obtain multi-scale features in the network and improves the detection precision of the network by fusing the high-resolution and the semantic information. The structure is shown in Figure 5.

In the process of constructing pyramid feature mapping, the output features of the second stage to the last residuals in the fifth stage of the backbone network are reduced by a 1 × 1 convolution operation to obtain different scale features as {C2, C3, C4, C5}. Then, they are connected by top-down and horizontal connections to form pyramid features {P2, P3, P4, P5}. The convolution operation of 1 × 1 is to reduce the number of convolution kernels, that is, to reduce the number of channels of the feature maps without changing the size of the feature maps.

The human visual system often does not understand and process all information. Instead, it focuses attention on some significant or interesting information, which helps to filter out unimportant information and improve the efficiency of information processing. To make rational use of the limited visual information processing resources, humans select and focus on specific parts of the visual area. This visual processing mechanism is called the saliency mechanism [29,30,31]. In detection tasks, extracting the detailed information of a specific area is the key to improving detection efficiency. The saliency mechanism can select the focus position in the input information of an image, which makes the detection network pay more attention to the more significant feature information in the input data so that the features extracted by the network are more distinguishable. In this paper, the saliency module is integrated after the feature pyramid module. Through the saliency mechanism, the number of false detections caused by background information can be reduced, thereby improving the detection precision of the network.

In the saliency module is shown in Figure 6, a malignant tumor image is taken as an example. The spiking CNN is employed to extract the features. Then, the two-dimensional feature maps generated by the spike convolution layer are summed and the mask is calculated to obtain the saliency feature map.

## 4. Datasets

In this paper, two datasets are used to verify the proposed model, namely the dataset of breast ultrasound images [32] and the DDSM database [33,34]. Because the dataset does not contain labels for object detection, the labels are manually labeled using the open-source script LabelImg (https://github.com/tzutalin/labelImg, accessed on 20 October 2022) on GitHub.

### 4.1. Breast Ultrasound Images

Ultrasound scanning is mainly used for breast cancer detection and early detection. In addition, it is safe compared to other radiographic imaging techniques. This dataset is collected from breast ultrasound images of 600 female patients between 25 and 75 years old and contains 780 images. The average size of images is 500 × 500 pixels. The images are divided into three categories: normal, benign, and malignant. The images are in the PNG format.

The three types of images in the dataset are shown in Figure 7. Figure 7a is a normal image, Figure 7b is a benign tumor image, and Figure 7c is a malignant tumor image. The number of images in each category is shown in Table 2. As shown in Table 2, the dataset contains 133 normal images, 437 benign images, and 210 malignant images. Since the experiment does not involve normal instances, we increase the number of malignant instances and realize data expansion by rotating the malignant image 90 degrees. Finally, 420 malignant images are obtained.

### 4.2. DDSM Dataset

The digital database for screening mammography (DDSM) is a digital film-screen mammography database containing relevant ground truth and other information. The database contains 2620 4-view mammography screening examinations. Figure 8 shows some cases with unusual attributes.

The four standard views of each case were digitized in one of four different views. Table 3 shows some of the characteristics of these scanners and provides a calibration equation for converting pixel values to optical density.

According to the severity of the findings, the cases are divided into different volumes. The normal volume contains mammograms for screening examinations; these examinations are considered normal, and a normal screening examination was performed four years later (plus or minus six months). The amount of benign non-revised visits includes abnormalities in the examination, which is worth noting but does not require any additional examinations. Benign tumors include some suspicious cases. The patient was recalled for some additional tests, and benign tumors were found. The cancer volume contains histologically confirmed cases of cancer. Each volume may contain some cases, in addition to more serious findings that led to the assignment of cases to a particular volume, but also less serious findings. Table 4 shows the breakdown of 2620 mammography equipment and volume types in the database.

Each case in the DDSM includes the age of patients, the date of the screening examination, the date the mammogram was digitized, and the ACR breast density assigned by the radiologist. Except for the normal volume, all cases in the volume contain pixel-level abnormal ground truth labels.

## 5. Experimental Results

The datasets used in this work can be applied to the segmentation, classification, and detection of breast cancer. The data provides classification labels and segmentation labels. However, the datasets do not contain labels for object detection. Therefore, the labels are manually labeled using the open-source script LabelImg. An example of the annotated image is shown in Figure 9. Figure 9a is the annotation of malignant lesions in an ultrasound image, and Figure 9b is the annotation of benign lesions in the DDSM database.

### 5.1. Experiment Settings

The parameters in the network are set according to experience, as shown in Table 5. In the training process, the iteration includes eight groups, and these samples are divided a further eight times to participate in the network training. Therefore, the value of batch size is set to 64, and the subdivision is set to 8. The momentum is set to 0.9, the value of decay is set to 0.0005, and the learning rate is set to 0.001. Here, $V_{rest}$ is the membrane potential of neurons in a resting state. In this paper, it is set to 0 mV. Addionally, $V_{threshold}$ is the threshold that determines the spike firing or not.

### 5.2. Breast Ultrasound Dataset

The dataset of breast ultrasound images is categorized into three classes, i.e., normal, benign, and malignant, as shown in Figure 7. In our work, the dataset combines normal and benign into negative and classifies malignant as positive.

To verify and analyze the performance of the proposed methods on the dataset of breast ultrasound images, the effects of the presence or absence of feature pyramid network and saliency module are investigated. As shown in Table 6, the precision of the SNN backbone for detecting benign and malignant lesions is 93.18% and 71.31%, respectively. Furthermore, the value of mean average precision (mAP) is 82.25%. The recall of benign and malignant lesions is 94.12% and 79.10%, respectively. When FPN is used for SNN, the network can achieve a mAP value of 85.69%. The recall of benign and malignant lesions is 96.83% and 78.11%, respectively. When the saliency module is used for SNN, the network can achieve a mAP value of 84.62%. The recall of benign and malignant lesions is 96.38% and 77.11%, respectively. When both the FPN and saliency module are applied for SNN, this work achieves a remarkable performance of 90.67% on the dataset of breast ultrasound images. The recall of benign and malignant lesions is 98.19% and 87.56%, respectively. It can be seen that the SNN backbone achieves the lowest mAP value, and the combination of SNN and FPN or an saliency module can improve the detection precision.

Figure 10 is a schematic diagram of the detection results. Figure 10a is the detection result of a benign lesion, and Figure 10b is the detection result of a malignant lesion. The method proposed in this paper can accurately detect the type and location of the lesion on the breast ultrasound dataset.

Detection results comparison of different algorithms on the breast ultrasound dataset is shown in Table 7. The method of SSD provides 81.64% mAP. The detection precision of SSD for benign and malignant lesions is 92.19% and 71.08%, respectively. The mAP of 80.27% is achieved by YOLOv1, and the detection precision of YOLOv1 for benign and malignant lesions is 93.91% and 66.64%, respectively. The mAP of 80.86% is obtained by using YOLOv2, and the detection precision of YOLOv2 for benign and malignant lesions is 90.08% and 71.63%, respectively. Using YOLOv3 provides 81.73% mAP, and the detection precision of YOLOv3 for benign and malignant lesions is 93.78% and 69.68%, respectively. Furthermore, YOLO-Tiny provides 75.69% mAP, and the detection precision of YOLO-Tiny for benign and malignant lesions is 93.78% and 69.68%, respectively. The YOLO-Lite [35] can achieve 72.25% mAP, and the detection precision of YOLO-Lite for benign and malignant lesions is 90.53% and 53.96%, respectively. Our work can achieve 90.67% mAP, and the detection precision of benign and malignant lesions is 96.61% and 84.72%, respectively. It can be seen that our work is superior to other networks.

Compared with ANNs, a theoretical advantage of SNN is that it can save computing time. Therefore, this paper compares the computing time performance of several models on a single image, as shown in Table 8. Table 8 compares the computing time of different models on CPU and GPU. The performance of different task scenarios and models is often different. For simple task scenarios, simple models often perform better than complex models. It can be seen that the computing time of SSD on CPU and GPU is 1900 ms and 910 ms, respectively. The computing time of YOLOv1 on CPU and GPU is 1752 ms and 901 ms, respectively. The computing time of YOLOv2 on CPU and GPU is 1301 ms and 730 ms, respectively. The computing time of YOLOv3 on CPU and GPU is 800 ms and 42 ms, respectively. Here, YOLO-Lite consumes the least time, and the computing time on CPU and GPU is 141 ms and 16 ms, respectively. The second fastest is the YOLO-Tiny model. The computing time on CPU and GPU is 172 ms and 20 ms, respectively. The YOLO-Tiny and YOLO-Lite are two lightweight models, so they consume the least time, but the detection results are not as good as other models. The model proposed in this paper is optimal under the trade-off between computing time and precision.

### 5.3. DDSM Dataset

This work writes the path of all the LJPEG suffix files in the dataset to a temporary text. Then it reads the text line by line, loads the corresponding LJPEG file according to the path each time, and reads the information in the corresponding ‘ics’ format file under the path at the same time, before finally converting the LJPEG file to a JPG format. The schematic diagram of the converted image in DDSM is shown in Figure 8.

To verify and analyze the performance of the proposed methods on the DDSM dataset, the effects of the presence or absence of the feature pyramid network and saliency module are investigated. As shown in Table 9, the precision of the SNN backbone for detecting benign and malignant lesions is 75.52% and 90.29%, respectively. Furthermore, the value of mAP is 82.90%. The recall of the SNN backbone for benign and malignant lesions is 95.96% and 99.0%, respectively. When FPN is used for SNN, the precision for detecting benign and malignant lesions is 73.60% and 90.64%, respectively, and the network can achieve a mAP value of 84.98%. The recall of benign and malignant lesions is 96.46% and 99.50%, respectively. When the saliency module is used for SNN, the precision for detecting benign and malignant lesions is 77.16% and 91.31%, respectively, and the network can achieve a mAP value of 84.23%. The recall of benign and malignant lesions is 97.98% and 99.50%, respectively. When both FPN and the saliency module are applied for SNN, this work achieves a remarkable performance of 92.81% on the DDSM dataset. The recall of benign and malignant lesions is 98.99% and 99.50%, respectively. It can be seen that the SNN backbone achieves the lowest mAP value, and the combination of SNN and FPN (or the saliency module) can improve the detection precision.

Figure 11 is a schematic diagram of the detection results on the DDSM dataset. Figure 11a is the detection result of a benign lesion, while Figure 11b is the detection result of a malignant lesion. Figure 11c contains one benign tumor; however, the model mistakenly detects it as containing two benign tumors and one malignant tumor.

The detection results comparison of different algorithms on the DDSM dataset is shown in Table 10. The method of SSD provides 80.94% mAP. The detection precision of SSD for benign and malignant lesions is 70.26% and 91.62%, respectively. The mAP of 74.92% is achieved by YOLOv1, and the detection precision of YOLOv1 for benign and malignant lesions is 62.88% and 86.95%, respectively. The mAP of 75.58% is obtained by using YOLOv2, and the detection precision of YOLOv2 for benign and malignant lesions is 66.60% and 84.56%, respectively. Using YOLOv3 provides 77.94% mAP, and the detection precision of YOLOv3 for benign and malignant lesions is 67.92% and 87.96%, respectively. Here, YOLO-Tiny provides 66.46% mAP, and the detection precision of YOLO-Tiny for benign and malignant lesions is 63.46% and 69.47%, respectively. The YOLO-Lite can achieve 67.66% mAP, and the detection precision of YOLO-Lite for benign and malignant lesions is 51.96% and 83.35%, respectively. Our work can archive 92.81% mAP, and the detection precision of benign and malignant lesions is 89.51% and 96.11%, respectively. It can be seen that our work is superior to other networks on the DDSM dataset.

Table 11 compares the CPU and GPU computing time of different models on the DDSM dataset. It can be seen that the computing time of SSD on CPU and GPU is 2100 ms and 1310 ms, respectively. Here, YOLO-Lite consumes the least time, and the computing time on CPU and GPU is 401 ms and 107 ms, respectively. The second fastest is the YOLO-Tiny model, which takes 475 ms and 120 ms on CPU and GPU, respectively. However, the detection results of these two models are not as good as other models. Considering the trade-off between computing time and precision, the performance of the proposed model on the DDSM dataset is superior to other models.

### 5.4. Energy Efficiency

As mentioned above, SNN has low power consumption. This section analyzes the energy consumption of a Spiking-YOLO model. Since YOLO-Tiny is a lightweight network, it consumes the least energy in the YOLO series. To highlight the advantage of our work in energy consumption, Spiking-YOLO is compared with YOLO-Tiny. Table 12 shows the energy comparison results of Spiking-YOLO and YOLO-Tiny.

## 6. Conclusions

In this paper, SNN is used in the field of object detection based on medical images for the first time. This work relies on the YOLO framework and uses the feature pyramid structure to obtain the multi-scale features of the image. By fusing the high resolution of low-level features and the strong semantic information of high-level features, the detection precision of the network is improved. The spatial and channel saliency modules are employed to improve the performance. Due to the fact that SNN cannot be trained using the backpropagation algorithm directly, a method of converting DNN to SNN is proposed. The theoretical proof is then given. The detection results of our method are superior to other models both on breast ultrasound and DDSM datasets. However, the detection performance of malignant tumors is lower than that of benign tumors on breast ultrasound images. The detection performance of malignant tumors is higher than that of benign tumors on the DDSM dataset. Future work will improve the performance and will allow us to apply SNN for object detection of different modalities based on medical images.

## Figures and Tables

**Figure 1 entropy-24-01543-f001:**
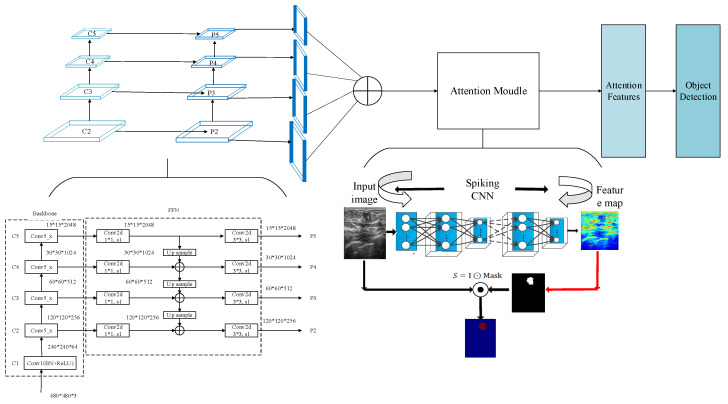
Object detection network with a multi-scale saliency fusion model.

**Figure 2 entropy-24-01543-f002:**
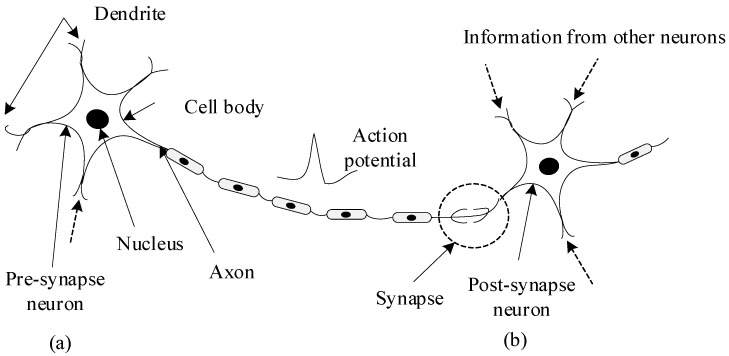
The generation and transmission of spikes. (**a**) a pre-synaptic neuron; (**b**) a post-synaptic neuron.

**Figure 3 entropy-24-01543-f003:**
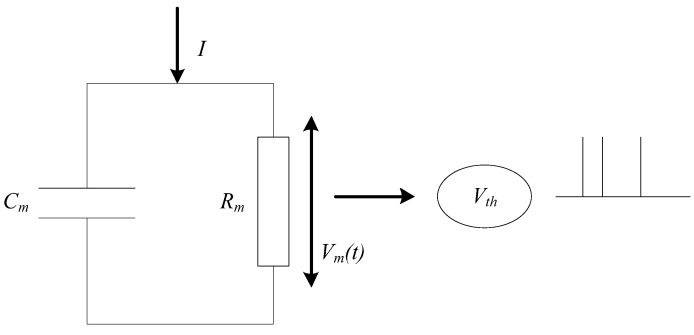
Circuit schematic of the LIF model.

**Figure 4 entropy-24-01543-f004:**
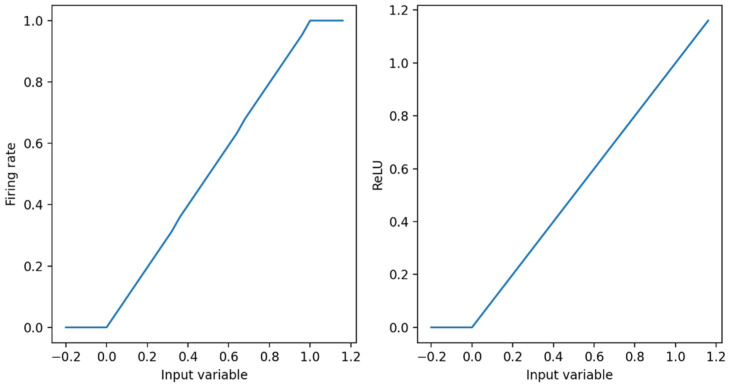
Output curve of spike neuron and ReLu activation function.

**Figure 5 entropy-24-01543-f005:**
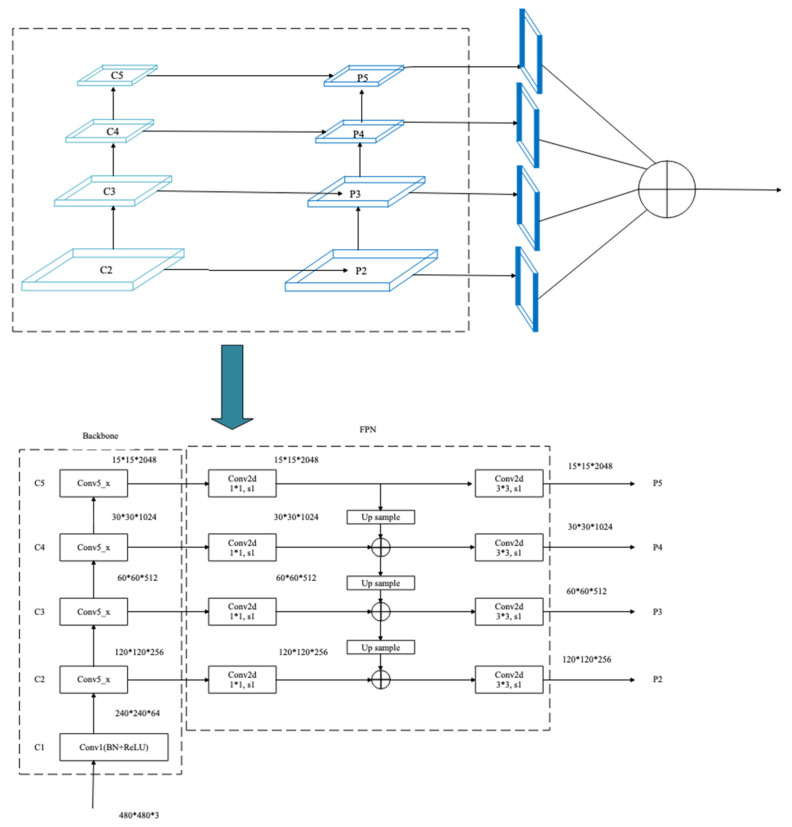
Schematic diagram of the feature pyramid structure. Input data take breast ultrasound images as an example.

**Figure 6 entropy-24-01543-f006:**
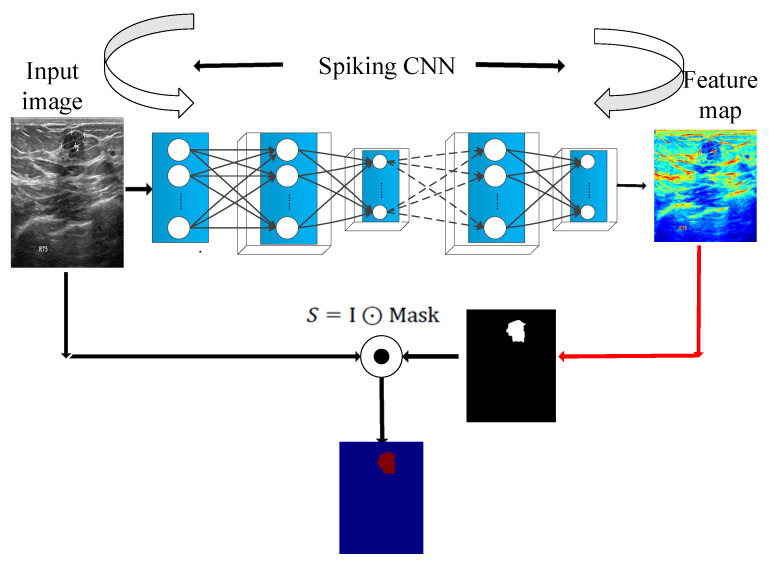
Saliency-based feature extraction using the spiking CNN.

**Figure 7 entropy-24-01543-f007:**
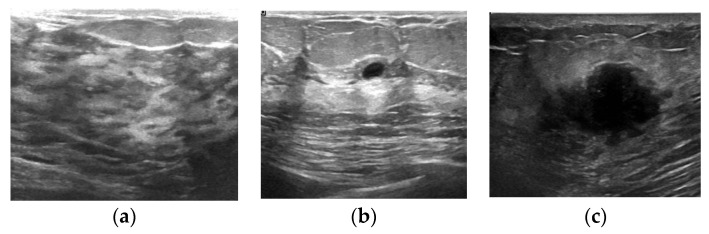
Three types of images in the dataset of breast ultrasound images. (**a**) Normal; (**b**) benign; (**c**) malignant.

**Figure 8 entropy-24-01543-f008:**
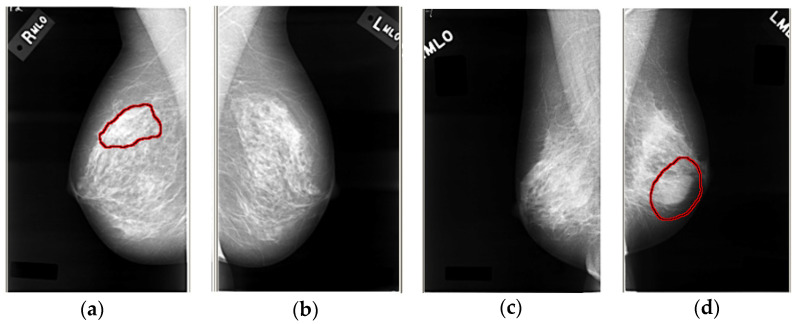
Some cases with unusual attributes. Here, (**a**,**b**) is the left and right breast of patients with a malignant tumor, while (**c**,**d**) is the left breast and right breast of patients with a benign tumor.

**Figure 9 entropy-24-01543-f009:**
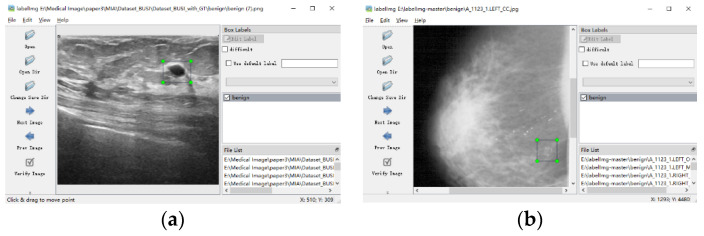
An example of annotated image. (**a**) Annotation of malignant lesions in an ultrasound image. (**b**) Annotation of benign lesions in the DDSM database.

**Figure 10 entropy-24-01543-f010:**
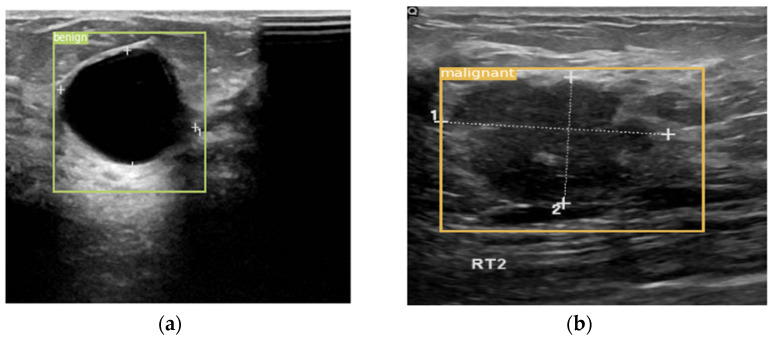
An example of a detection result. (**a**) The detection result for the benign lesion. (**b**) The detection result for the malignant lesion.

**Figure 11 entropy-24-01543-f011:**
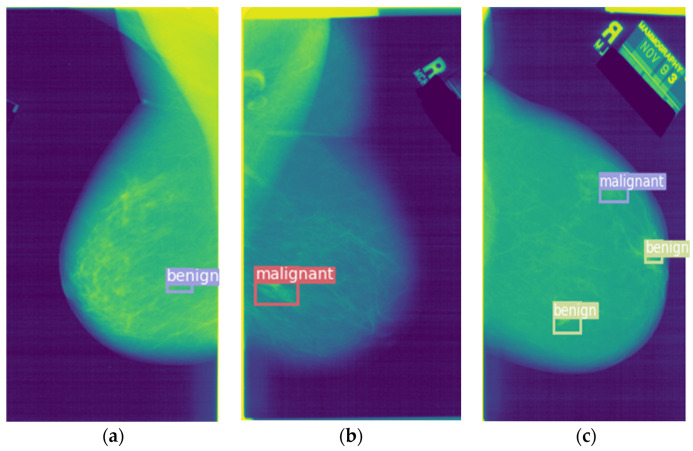
Schematic diagram of the detection result. (**a**) The detection result of a benign lesion. (**b**) The detection result of a malignant lesion. (**c**) Misdiagnosis of a benign tumor.

**Table 1 entropy-24-01543-t001:** Summary of deep learning methods in the field of object recognition.

DNNs	Classifiers	Regressors	Candidate Box Extraction	Backbone
R-CNN	SVM	SVM	Selective search	CNN
Fast R-CNN	Softmax	Linear regression	Selective search	CNN
Faster R-CNN	Softmax	Linear regression	Region proposal network	CNN
SSD	Softmax	Bounding box regression	-	VGG-16
YOLOv5	Logistic	DIOU_NMS	-	DSPDarkNet53

**Table 2 entropy-24-01543-t002:** The number of images in each class.

Classes of Images	Number of Images
Normal	133
Benign	437
Malignant	210
Total	780

**Table 3 entropy-24-01543-t003:** The sampling rate, number of gray scales, and the formula for estimating the optical density (OD) of each scanner from the gray value (GV) of mammograms used to digitize DDSM.

Digitizer	Sampling Rate (Microns)	Gray Levels (Bits)	Optical Density Calibration Equation
DBA M2100 ImageClear	42	16	OD = 4.26700423014133 + (−0.90303289757264) ∗ log10(GV)
Howtek 960	43.5	12	OD = 3.78928997845071 + (−0.00094568009377) ∗ GV
Lumisys 200 Laser	50	12	OD = 4.05977749300340 + (−0.00099080941710) ∗ GV
Howtek MultiRad850	43.5	12	OD = 3.96604095240593 + (−0.00099055807612) ∗ GV

**Table 4 entropy-24-01543-t004:** Contents of the DDSM database in the case.

Institution	Digitizer	Number of Cases by Most Severe Finding				Total
		Normal	Benign without Callback	Benign	Malignant	
MGH	DBA M2100 ImageClear	430	0	0	97	527
	Howtek 960	78	0	446	323	847
WFU	Lumisys 200 Laser	82	93	126	159	460
SH		0	48	202	234	484
WU	Howtek MultiRad850	105	0	96	101	302
Total		695	141	870	914	2620

**Table 5 entropy-24-01543-t005:** The parameters of neuron model and network for the experiments.

Neuron Model		Network	
Parameters	Value	Parameters	Value
$V_{rest}$	0 mV	Batch	64
$V_{threshold}$	1.0 mV	Subdivision	8
$V_{reset}$	0 mV	Momentum	0.9
τ	10 ms	Decay	0.0005
$\tau_{refractory}$	1 ms	Learning rate	0.001
Δt	0.01 ms	Ignore threshold	0.5

**Table 6 entropy-24-01543-t006:** The performance of SNN with different models on the breast ultrasound images.

Models	Precision (%)		Recall (%)		mAP (%)
	Benign	Malignant	Benign	Malignant	
SNN backbone	93.18	71.31	94.12	79.10	82.25
SNN with FPN	95.93	75.46	96.83	78.11	85.69
SNN with saliency module	95.39	73.86	96.38	77.11	84.62
This work	96.61	84.72	98.19	87.56	90.67

**Table 7 entropy-24-01543-t007:** Comparison of detection performance with different models on the breast ultrasound images.

Models	Precision (%)		mAP (%)	Recall (%)
	Benign	Malignant		
SSD	92.19	71.08	81.64	76.83
YOLOv1	93.91	66.64	80.27	74.55
YOLOv2	90.08	71.63	80.86	76.76
YOLOv3	93.78	69.68	81.73	76.29
YOLO-Tiny	74.97	76.41	75.69	76.78
YOLO- Lite	90.53	53.96	72.25	68.29
YOLOv5	94.31	76.60	85.46	80.75
This work	96.61	84.72	90.67	86.80

**Table 8 entropy-24-01543-t008:** Time consumption of different models on the dataset of breast ultrasound images.

Models	Time Consumption	
	CPU	GPU
SSD	1900 ms	910 ms
YOLOv1	1752 ms	901 ms
YOLOv2	1301 ms	730 ms
YOLOv3	800 ms	42 ms
YOLO-Tiny	172 ms	20 ms
YOLO-Lite	141 ms	16 ms
This work	720 ms	37 ms

**Table 9 entropy-24-01543-t009:** The performance of SNN with different models on the DDSM dataset.

Models	Precision (%)		Recall (%)		mAP (%)
	Benign	Malignant	Benign	Malignant	
SNN backbone	75.52	90.29	95.96	99.0	82.90
SNN with FPN	73.60	90.64	96.46	99.50	84.98
SNN with saliency module	77.16	91.31	97.98	99.50	84.23
This work	89.51	96.11	98.99	99.50	92.81

**Table 10 entropy-24-01543-t010:** Comparison of detection performance with different models on the DDSM.

Models	Precision (%)		mAP (%)	Recall (%)
	Benign	Malignant		
SSD	70.26	91.62	80.94	88.77
YOLOv1	62.88	86.95	74.92	81.95
YOLOv2	66.60	84.56	75.58	80.26
YOLOv3	67.92	87.96	77.94	84.17
YOLO-Tiny	63.46	69.47	66.46	66.21
YOLO- Lite	51.96	83.35	67.66	74.63
YOLOv5	87.21	97.69	92.45	97.27
This work	89.51	96.11	92.81	95.60

**Table 11 entropy-24-01543-t011:** Time consumption of different models on the DDSM dataset.

Models	Time Consumption	
	CPU	GPU
SSD	2100 ms	1310 ms
YOLOv1	1964 ms	980 ms
YOLOv2	1600 ms	851 ms
YOLOv3	900 ms	276 ms
YOLO-Tiny	475 ms	120 ms
YOLO-Lite	401 ms	107 ms
This work	866 ms	267 ms

**Table 12 entropy-24-01543-t012:** Comparison results of energy consumption between Spiking-YOLO and YOLO-Tiny.

Models	Calculation Form	MAC (pJ)	AC (pJ)	FLOPs	Energy (J)
Tiny-YOLO	32-bit FL	4.6	/	$6.97\times 10^{9}$	0.032
	32-bit INT	3.2	/		0.022
Spiking-YOLO	32-bit FL	/	0.9	$5.28\times 10^{7}$	$4.75\times 10^{-5}$
	32-bit INT	/	0.1		$5.28\times 10^{-6}$

## Data Availability

The datasets used and analyzed during the current study are available from the corresponding author on reasonable request.

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
