# Peer review of "Spiking Neural Network Based on Multi-Scale Saliency Fusion for Breast Cancer Detection"

_entropy, 2022, doi:10.3390/e24111543_

Round 1
Reviewer 1 Report
In this paper, the authors proposed a SNN method to the field of object detection based on medical images using the method of converting DNN to SNN. The proposed method is applied to detect the location and classification of breast lesions with ultrasound and X-ray datasets, and the results are satisfactory.
There are some points which need to be cleared or included in their paper by the authors.
1. Some of the abbreviations are repeated, i.e. DNN and ANN. Suggestion is not to abbreviate in abstract.
2. As the dataset is imbalance (Table 1, normal instances are 133 while benign are 437), the authors should try to generate synthetic data to reduce the unbalancing in the datasets.
3. Give details of DDSM in tabular form.
4. In Tables 6 and 9, the recall should be added as recall is the better model evaluation for health type applications.
Author Response
This document outlines the additional information provided in addressing the reviewers’ comments. The following numbering list clearly outlines how we have addressed all the comments raised by each of the two reviewers. The changes to manuscript are also highlighted in the main paper using red coloured text.
We wish to express sincere thanks to the editors and reviewers for their comprehensive, detailed and insightful comments. We have taken the observations on board and we feel that this input has greatly added to the quality of the paper.
Reviewer 1
--------------------------------
- Some of the abbreviations are repeated, i.e. DNN and ANN. Suggestion is not to abbreviate in abstract.
[Author Response]: Thank you very much for your suggestion. This suggestion has now been addressed in the revised paper.
[Author Action]: We updated the manuscript in the revised paper. The following text is the revised content in abstract.
“Deep neural networks have been successfully applied in the field of image recognition and object detection, and the recognition results are close to or even beyond human beings. Deep neural network takes the activation function as the basic unit. It is inferior to the spiking neural network which takes the spiking neuron model as the basic unit in the aspect of biological interpretability. Spiking neural network is considered as the third generation artificial neural network, which is event-driven and has low power consumption. It modulates the process of nerve cells from receiving stimulus to firing spikes. However, it is difficult to train spiking neural network directly due to the non-differentiable spiking neurons. In particular, it is impossible to train spiking neural network using the back-propagation algorithm directly. Therefore, the application scenarios of spiking neural network are not as extensive as deep neural network, and spiking neural network is mostly used in simple image classification tasks. This paper proposed a spiking neural network method to the field of object detection based on medical images using the method of converting deep neural network to spiking neural network. The detection framework relies on the YOLO structure and uses the feature pyramid structure to obtain the multi-scale features of the image. By fusing the high resolution of low-level features and the strong semantic information of high-level features, the detection precision of the network is improved. The proposed method is applied to detect the location and classification of breast lesions with ultrasound and X-ray datasets, and the results are 90.67% and 92.81 respectively.”
- As the dataset is imbalance (Table 1, normal instances are 133 while benign are 437), the authors should try to generate synthetic data to reduce the unbalancing in the datasets.
[Author Response]: Thank you very much for pointing out such a constructive suggestion. This suggestion has now been addressed in the revised paper. In fact, we considered the problem of data imbalance, but did not describe it in the article. Since the experiment did not involve normal instances, we enhanced the data of malignant instances and realized data expansion by rotating the malignant image 90 degrees. Finally, 420 malignant images are obtained.
[Author Action]: We updated the manuscript in the revised paper. The following text is the revised content in the second paragraph of Section 4.1.
“As shown in table 2, the data set contains 133 normal images, 437 benign images, and 210 malignant images. Since the experiment does not involve normal instances, we increase the number of malignant instances and realize data expansion by rotating the malignant image 90 degrees. Finally, 420 malignant images are obtained.”
- Give details of DDSM in tabular form.
[Author Response]: Thank you very much for your suggestion. This suggestion has now been addressed in the revised paper.
[Author Action]: We updated the manuscript in the revised paper. The following text is the revised content in Section 4.2.
- In Tables 6 and 9, the recall should be added as recall is the better model evaluation for health type applications.
[Author Response]: Thank you very much for pointing out such a constructive suggestion. This suggestion has now been addressed in the revised paper by adding recall.
[Author Action]: We updated the manuscript in the revised paper. The following text is the revised content in Tables 7 and 10.

Reviewer 2 Report
This is an interesting paper. There are minor concerns to be addressed.
1) Each method for artificial intelligence should be summarized as main table for the better understanding.
2) The key findings should be addressed in the abstract.
3) "Related works" seems redundant. It would be better to be placed as Supplementary materials.
4) As an outcome measure, please, show the AUROC.
5) How was mAP calculated?
6) The study finding has a clinical implication for other disease where ultrasonography is the mainstay for surveillance (e.g. chronic liver disease). It should be addressed.
7) In the similar context, reference (PMID 31726817) should be cited together.
Author Response
This document outlines the additional information provided in addressing the reviewers’ comments. The following numbering list clearly outlines how we have addressed all the comments raised by each of the two reviewers. The changes to manuscript are also highlighted in the main paper using red coloured text.
We wish to express sincere thanks to the editors and reviewers for their comprehensive, detailed and insightful comments. We have taken the observations on board and we feel that this input has greatly added to the quality of the paper.
- Each method for artificial intelligence should be summarized as main table for the better understanding.
[Author Response]: Thank you very much for your suggestion. This suggestion has now been addressed in the revised paper.
[Author Action]: We updated the manuscript in the revised paper. The following text is the revised content in Section 2.
- The key findings should be addressed in the abstract.
[Author Response]: Thank you very much for pointing out such a constructive suggestion. This suggestion has now been addressed in the revised paper.
[Author Action]: We updated the manuscript in the revised paper. The following text is the revised content in the abstract.
“Deep neural networks have been successfully applied in the field of image recognition and object detection, and the recognition results are close to or even beyond human beings. Deep neural network takes the activation function as the basic unit. It is inferior to the spiking neural network which takes the spiking neuron model as the basic unit in the aspect of biological interpretability. Spiking neural network is considered as the third generation artificial neural network, which is event-driven and has low power consumption. It modulates the process of nerve cells from receiving stimulus to firing spikes. However, it is difficult to train spiking neural network directly due to the non-differentiable spiking neurons. In particular, it is impossible to train spiking neural network using the back-propagation algorithm directly. Therefore, the application scenarios of spiking neural network are not as extensive as deep neural network, and spiking neural network is mostly used in simple image classification tasks. This paper proposed a spiking neural network method to the field of object detection based on medical images using the method of converting deep neural network to spiking neural network. The detection framework relies on the YOLO structure and uses the feature pyramid structure to obtain the multi-scale features of the image. By fusing the high resolution of low-level features and the strong semantic information of high-level features, the detection precision of the network is improved. The proposed method is applied to detect the location and classification of breast lesions with ultrasound and X-ray datasets, and the results are 90.67% and 92.81 respectively.”
- "Related works" seems redundant. It would be better to be placed as Supplementary materials.
[Author Response]: Thank you very much for your suggestion. Because our work focus on object detection, we explain some mainstream algorithms of object detection in related work. Thus explaining why we propose new algorithms. After careful consideration by our team, we have decided that related work should be introduced. Thank you very much.
- As an outcome measure, please, show the AUROC.
[Author Response]: Thank you very much for your suggestion. In object detection tasks, mAP is a common measurement standard. AUC is also important, however, AUC is usually used in classification tasks. In order to compare with other models, we choose the same indicators for comparison, so AUC is not included. To further prove the performance of the model, as shown in tables 7 and 10, we add the calculation of recall.
- How was mAP calculated?
[Author Response]: Thank you very much for your suggestion. mAP can be obtained through the following code:
def mAP():
detpath,annopath,imagesetfile,cachedir,class_path = get_dir('kitti')
ovthresh=0.3,
use_07_metric=False
rec = 0; prec = 0; mAP = 0
class_list = get_classlist(class_path)
for classname in class_list:
rec, prec, ap = voc_eval(detpath,
annopath,
imagesetfile,
classname,
cachedir,
ovthresh=0.5,
use_07_metric=False,
kitti=True)
print('on {}, the ap is {}, recall is {}, precision is {}'.format(classname, ap, rec[-1], prec[-1]))
mAP += ap
mAP = float(mAP) / len(class_list)
return mAP
- The study finding has a clinical implication for other disease where ultrasonography is the mainstay for surveillance (e.g. chronic liver disease). It should be addressed.
[Author Response]: Thank you very much for your suggestion. Chronic liver disease is a problem worth paying attention to. Because our work mainly focuses on the detection of breast tumors, we do not pay attention to other diseases at present. This is also the inadequacy of our research, and we will focus on this research in future work.
- In the similar context, reference (PMID 31726817) should be cited together.
[Author Response]: Thank you very much for your suggestion. This suggestion has now been addressed in the revised paper.
[Author Action]: We updated the manuscript in the revised paper. The 7th reference is (PMID 31726817).

Round 2
Reviewer 1 Report
The paper is acceptable in current form.